# Identification of Unique Genetic Biomarkers of Various Subtypes of Glomerulonephritis Using Machine Learning and Deep Learning

**DOI:** 10.3390/biom12091276

**Published:** 2022-09-10

**Authors:** Jianbo Qing, Fang Zheng, Huiwen Zhi, Hasnaa Yaigoub, Hasna Tirichen, Yaheng Li, Juanjuan Zhao, Yan Qiang, Yafeng Li

**Affiliations:** 1The Fifth Clinical Medical College, Shanxi Medical University, Taiyuan 030001, China; 2Department of Nephrology, Shanxi Provincial People’s Hospital (Fifth Hospital), Shanxi Medical University, Taiyuan 030001, China; 3College of Information and Computer, Taiyuan University of Technology, Taiyuan 030001, China; 4Institutes of Biomedical Sciences, Shanxi University, Taiyuan 030001, China; 5Laboratory for Molecular Diagnosis and Treatment of Kidney Disease, Shanxi Provincial People’s Hospital (Fifth Hospital), Shanxi Medical University, Taiyuan 030001, China; 6Core Laboratory, Shanxi Provincial People’s Hospital (Fifth Hospital), Shanxi Medical University, Taiyuan 030001, China; 7Shanxi Provincial Key Laboratory of Kidney Disease, Taiyuan 030001, China; 8Academy of Microbial Ecology, Shanxi Medical University, Taiyuan 030001, China

**Keywords:** glomerulonephritis, immune-related genes, immune infiltration, machine learning, deep learning

## Abstract

(1) Objective: Identification of potential genetic biomarkers for various glomerulonephritis (GN) subtypes and discovering the molecular mechanisms of GN. (2) Methods: four microarray datasets of GN were downloaded from Gene Expression Omnibus (GEO) database and merged to obtain the gene expression profiles of eight GN subtypes. Then, differentially expressed immune-related genes (DIRGs) were identified to explore the molecular mechanisms of GN, and single-sample gene set enrichment analysis (ssGSEA) was performed to discover the abnormal inflammation in GN. In addition, a nomogram model was generated using the R package “glmnet”, and the calibration curve was plotted to evaluate the predictive power of the nomogram model. Finally, deep learning (DL) based on a multilayer perceptron (MLP) network was performed to explore the characteristic genes for GN. (3) Results: we screened out 274 common up-regulated or down-regulated DIRGs in the glomeruli and tubulointerstitium. These DIRGs are mainly involved in T-cell differentiation, the RAS signaling pathway, and the MAPK signaling pathway. ssGSEA indicates that there is a significant increase in DC (dendritic cells) and macrophages, and a significant decrease in neutrophils and NKT cells in glomeruli, while monocytes and NK cells are increased in tubulointerstitium. A nomogram model was constructed to predict GN based on 7 DIRGs, and 20 DIRGs of each subtype of GN in glomeruli and tubulointerstitium were selected as characteristic genes. (4) Conclusions: this study reveals that the DIRGs are closely related to the pathogenesis of GN and could serve as genetic biomarkers in GN. DL further identified the characteristic genes that are essential to define the pathogenesis of GN and develop targeted therapies for eight GN subtypes.

## 1. Introduction

Glomerulonephritis (GN) is an important public health problem worldwide that can cause end-stage renal disease (ESRD) [1]. It affects people of all regions and all ages, is usually more prevalent among young people, and is difficult to treat [2]. GN can be divided into primary and secondary GN [3], and it can present in a variety of ways, but is usually accompanied by clinical features of hematuria, proteinuria, hypertension, and renal failure [3]. It is worth mentioning that GN is not only a single entity, but is potentially linked to many diseases and other systems [4]. GN diagnosis relies on kidney biopsy, the results of which separate glomerulonephritis into a variety of more specific pathologies.

Current treatments for GN focus on optimizing supportive therapy [5]. Immunosuppressants and monoclonal antibodies are not effective enough to prevent the development of GN [6], which further indicates the importance of early diagnosis and personalized treatment for the prevention and treatment of GN. Rapid advances in sequencing technologies revealed several disease-causing genes susceptibility loci and disease-causing genes for different subtypes of GN [7], which brings new light in the precise diagnosis and treatment of GN. Although there is abundant sequencing data of different GN subtypes, researchers performing conjoint analysis of these data are rare. As machine learning (ML) and deep learning (DL) are more used in genomics and provide promising results [8,9], they becomes more reliable to explore the unique genetic signatures of GN.

In this light, microarray data of several subtypes of GN were downloaded from the Gene Expression Omnibus (GEO) database [10] and merged to further identify differentially expressed genes in glomeruli and tubulointerstitium, in order to explore the molecular mechanism of GN. Additionally, abnormal immune infiltration was studied to investigate the immune microenvironment of GN. ML and DL were performed to identify genetic biomarkers of eight GN subtypes, which offers a molecular perspective to understand the molecular mechanisms and develop personalized treatment of GN.

## 2. Materials and Methods

### 2.1. Data Collection and Processing

To discover the potential signatures of different subtypes of GN, the keyword “glomerulonephritis” was used to search the gene expression profiles of GN in the GEO database. To reduce batch effects and improve the accuracy of the analysis, datasets containing multiple subtypes of GN were selected. Four datasets were finally filtered and downloaded, then log2 transformation and gene symbols conversion were performed for gene expression profiles using R language (version 4.1.0). To combine multiple datasets, the R software package “inSilicoMerging” (version 2.1.0) was used. Then, “combat” function was used to remove the batch effect of merged data [11]. The details of the datasets are presented in Table 1. Additionally, 1793 immune-related genes were obtained from the import database (https://www.immport.org/shared/, accessed on 5 July 2022) [12] (Appendix A).

### 2.2. Identification of Differentially Expressed Genes and Enrichment Analysis

The differentially expressed genes (DEGs) of glomeruli and tubulointerstitium were identified using “limma” package [15]. Genes with a *p* value < 0.05 were considered as significant DEGs. The significant DEGs of glomeruli and tubulointerstitium in GN were combined to identify the differentially expressed immune-related genes (DIRGs) with predictive and diagnostic significance. Furthermore, the package “clusterProfiler” (Version 3.14.3) was used for enrichment analysis to explore the function of genes based on the Kyoto Encyclopedia of Genes and Genomes (KEGG) and Gene Ontology (GO) analysis [16].

### 2.3. Immune Signatures of Different Subtypes of Glomerulonephritis

Multiple immune-cell-mediated inflammation is a crucial feature of GN. To explore the infiltration of immune cells in different subtypes of GN, R package “ImmuCellAI” was used to perform single-sample gene set enrichment analysis (ssGSEA), which simulated the flow cytometry process to predict 24 cell types abundance by hierarchical strategy. Thr 24 types of immune cells contain two layers: layer 1 consists of 10 cell types (DC, B-cell, monocyte, macrophage, NK, neutrophil, CD4 T, CD8 T, NKT, and Tgd), while layer 2 consists of 14 T-cell subtypes (CD4 naive, CD8 naive, Tc, Tex, Tr1, nTreg, iTreg, Th1, Th2, Th17, Tfh, Tcm, Tem, and MAIT) [17]. Additionally, the Wilcoxon rank sum test was used to compare the differences in immune cells between GN patients and healthy controls, and *p* values less than 0.05 were considered significant.

### 2.4. Machine Learning

The area under the receiver operating characteristic curve (AUC) of DIRGs was calculated using R package “pROC” (Version 1.17.0.1) [18]. The DIRGs with the top 20% of the AUC value both in glomeruli and tubulointerstitium were screened out for ML. Then, data 1 were employed for the training set, while data 2 were utilized as the test set, as the data in data 1 are more evenly distributed, and the main features of glomerulonephritis are in the glomeruli. The R package “glmnet” was used to obtain the DIRGs for optimal models and regression analysis was performed by LASSO-Cox function [19]. Furthermore, a nomogram model was established for the prediction and diagnosis of GN using the package “rms” [20], and the calibration curve was plotted to evaluate the predictive power of the nomogram model [21]. Additionally, the AUC value in the test set was calculated to evaluate the accuracy of the model.

### 2.5. Deep Learning

The biomarkers are critical to explore the pathogenesis, diagnosis, prognosis, and drug development of diseases. However, traditional ML methods are unable to classify multiple diseases and identify markers for each one of them. The characteristic genes of a disease can only be screened out by comparing the differences between a single disease and the control group, which results in a loss of meaningful information and a decrease in accuracy. Thus, to identify the unique biomarkers of each subtype of GN, a DL network was constructed based on multilayer perceptron (MLP) [22].

We constructed a 3-layer MLP network under the PyTorch framework [23], using 274 candidate genes that are commonly up-regulated or down-regulated in both glomeruli and tubulointerstitium as input data, and subtypes of GN as output results. First, (drop out value = 0.5), “batchnormal” function was used for the regularization of the network, and “ReLU” was selected as the activation function [24]. “Softmax” function was applied to achieve multi-classification and the “CrossEntropyLoss” function was used as the loss function when processing data [25].

After establishing the MLP network, a method called “Layerwise relevance propagation” (LRP) was used to evaluate the effect of DIRGs on the predicted results [26], and we scored the importance of each gene in each subtype of GN. Finally, the unique biomarkers of each subtype of GN were screened out according to the score of DIRGs.

## 3. Results

### 3.1. Data Processing

Microarray data from glomeruli and tubulointerstitium were combined into data 1 and data 2 (Appendix A), respectively. To ensure the accuracy and reliability of the results, we only retained GN with a sample size greater than 20. The information of data 1 and data 2 are shown in Table 2. After merging and removing batch effects, the data distribution of each dataset is consistent, and the UMAP plot shows the distribution characteristics of each data before and after removing batch effect (Figure 1). In addition, the boxplot was used to show the data distribution of each sample after removing the batch effect (Appendix A).

### 3.2. Identification of Differentially Expressed Genes

A total of 5510 DEGs between GN and the control group in data 1 are identified with *p* value < 0.05, including 3474 up-regulated and 2036 down-regulated genes. Also, 2338 up-regulated and 2441 down-regulated genes are obtained from data 2 (Figure 2A,B). In addition, there are 274 common differentially expressed IRGs (DIRGs) in glomeruli and tubulointerstitium (Figure 2C). Moreover, we found 170 common up-regulated and 104 common down-regulated DIRGs, which are considered candidate genes for ML and DL. In addition, 19 oppositely expressed DIRGs in the glomerulus and tubulointerstitium are also identified, and a log2FC–log2FC plot is used to visualize them (Figure 2D).

### 3.3. Enrichment Analysis

The enrichment analysis reveals the function of DIRGs in glomeruli and tubulointerstitium. In terms of the 170 common up-regulated DIRGs, they are related to phagosome; antigen processing and presentation; natural killer cell-mediated cytotoxicity; Th1, Th2, and Th17 cell differentiation; and T-cell receptor signaling pathway in KEGG enrichment analysis, while they are associated with innate immune response, the cytokine-mediated signaling pathway, and type I interferon signaling pathway (Figure 3A,B). Simultaneously, 104 common down-regulated DIRGs are mainly enriched in the ErbB signaling pathway, cytokine–cytokine receptor interaction, and the Ras signaling pathway (Figure 3C,D). We also performed enrichment analysis on the 19 oppositely expressed DIRGs and found that they are mainly involved in the NF-kappa B signaling pathway, TNF signaling pathway, and negative regulation of vascular-associated smooth muscle cell differentiation (Appendix A).

### 3.4. Immune Signatures of Different Subtypes of Glomerulonephritis

ssGSEA was performed on data 1 and data 2 to explore the abnormal immune infiltration in the glomeruli and tubulointerstitium of GN. The immune invasion of GN varies with subtypes and tissues (Figure 4A,B). From a holistic perspective, GN patients show a significant increase in DC and macrophage in glomeruli, although NK cells have a comparable level in both con and GN, but there is a significant decrease in neutrophils and NKT cell in glomeruli of GN, while monocytes and NK cells are increased in the tubulointerstitium of GN. (Figure 4C,D). There are significant changes in 14 T-cell subtypes: decreased iTreg, and increased MAIT, Tex, and Tem are observed in both glomeruli and tubulointerstitium (Figure 4E,F). Moreover, in terms of the eight GN subtypes, each of them has their own unique characteristics of immune infiltration (Figure 4G,H), which are the immune signatures that differentiate them from healthy kidney tissue.

### 3.5. Machine Learning

AUC values of 274 common up-regulated or down-regulated DIRGs in data 1 and data 2 were calculated using package “pROC” and are shown in the AUC–AUC plot (Figure 5A, Appendix A). To perform LASSO-Cox analysis, 11 DIRGs (*CX3CR1, TLR1, LYZ, TRIM27, HRG, LTB, LYN, CSHL1, TMSB10, ARG2, LTF*) with the top 20% AUC values both in glomeruli and tubulointerstitium were screened out as candidate genes. We used the package “glmnet” and performed 10-fold cross-validation to obtain the optimal model. Finally, 7 DIRGs (*ARG2, CSHL1, CX3CR1, LTF, LYZ, TMSB10, TRIM27*) were obtained with a criterion of λ(lambda) = 0.2 (Figure 5B,C).

A nomogram was constructed using data 1 as a training set based on seven DIRGs using logistic regression model. (Figure 6A). Then, a calibration curve was plotted to evaluate the predictive power of the nomogram model. When the risk of GN is between 0.5 and 0.7, the model predicts a slightly lower value than the actual, while the predicted value is higher than actual when the risk is between 0.7 and 0.8. Undoubtedly, actual risk and the predicted risk are very close in the nomogram model, suggesting it has high accuracy to predict and diagnose GN (Figure 6B). Additionally, the AUC value of 85.5935% further indicates that the nomogram model has excellent predictive power (Figure 6C).

### 3.6. Deep Learning

A MLP network containing three layers was constructed to classify eight subtypes of GN based on 274 candidate DIRGs from data 1 and data 2, and the scheme of the DL model is shown in Figure 7. With the increase in training times (x-axis), the change in loss value (y-axis) gradually decreases and finally converges around 14 in both the input data of glomeruli and tubulointerstitium (Figure 8A,B). Then, the “LRP” function was used to assess the importance score of each DIRGs for each subtype. Then, the top 20 DIRGs of each subtype of GN in glomeruli and tubulointerstitium were selected as characteristic genes (Appendix A), which can be considered as unique genetic biomarkers of the disease. Table 3 shows the top five DIRGs in glomeruli and tubulointerstitium of each subtype. Moreover, we used a confusion matrix to summarize the prediction results, and ROC curve to evaluate the accuracy of the DL model. The characteristic genes identified by DL have an AUC value of more than 0.8 for each subtypes of GN, in both the glomeruli and tubulointerstitium (Figure 8C–F), which indicates that eight subtypes of GN could be distinguished using 274 DIRGs and the characteristic genes may serve as unique genetic biomarkers for them.

## 4. Discussion

The incidence of GN is increasing worldwide, and although it could partly be related to changes in renal biopsy policy [27], it is certain that GN has become a vital part of renal disease due to the high risk of developing ESRD [28]. The diagnosis and treatment of GN has made great progress in the last 20 years, which benefits from the continuous exploration of molecular biology and pathophysiology. Exploring the genomics of kidney disease permits to better understand the link between pathophysiology and molecular function, and further accelerates advances in targeted therapies [29]. Genetic biomarkers are not only a bridge between clinical findings and molecular mechanisms of diseases, but also a sharp sword to achieve precision diagnosis and treatment.

To explore the characteristic genes and molecular mechanism of GN, two datasets containing glomeruli and tubulointerstitium sequencing data for eight subtypes of GN were downloaded and merged into data 1 and data 2. Further differential analysis and log2FC–log2FC comparison indicates that the DIRGs expression in glomeruli and tubulointerstitium of GN tends to be consistent. Although the pathophysiological changes of GN usually start from glomeruli, the glomeruli and tubulointerstitium are inseparable, sharing similar molecular mechanisms and, ultimately, leading to unique pathological changes [30]. Two hundred and seventy four common up-regulated or down-regulated DIRGs in glomeruli and tubulointerstitium represent the unique immunophenotypes of GN that could serve as candidate genes to explore the immune mechanism and genetic biomarkers of GN. Although renal biopsy provides the most accurate evidence for the precise diagnosis of GN, histopathological categories are not sufficient to explore the different molecular mechanisms of disease progression and response to treatment to achieve the most desired therapy [29]. The pathogenesis of renal impairment in GN can vary with different pathological types as well as clinical stages [31,32]. The mechanisms of GN are not very clear, but epigenetics and genetics are implicated in the pathogenesis [33]. Therefore, redefining diseases from the genetic level and tapping into the underlying immune mechanisms appears to be particularly important.

Different subtypes of GN have distinct molecular mechanisms, but there remains similar pathophysiological processes and immune mechanisms [34]. The KEGG enrichment analysis of 170 common up-regulated DIRGs indicates that the cytotoxicity mediated by NK cells and the differentiation of T-cells are crucial parts of the molecular mechanism of GN kidney inflammation. Numerous researchers have pointed out that NK cells can play an important role in renal injury [35], especially in inflammatory injury and fibrosis within the tubulointerstitial compartment [36]. In addition, the differentiation of Th1 and Th17 mediates the immune responses, and leads to proliferative and crescentic forms and renal inflammation in GN [37], while Th2 is mainly related to membranous patterns of injury [38]. The differentiation of Th1, Th2, and Th17, and the release of related cytokines, have a central role in inflammation and progression of kidney injury [39,40]. Regarding biological process, these 170 common up-regulated DIRGs in glomeruli and tubulointerstitium are significantly implicated in the type I interferon signaling pathway and interferon-gamma-mediated signaling pathway. Interferon (IFN) is related to the induction and progression of renal fibrosis [41]. Moreover, the 104 common down-regulated DIRGs are involved in the Ras signaling pathway, PI3K–Akt signaling pathway, MAPK signaling pathway, and others. The activity of RAS has many pathophysiologic functions in the progression of GN, and has become an important therapeutic target to improve proteinuria, hypertension, and other clinical symptoms [42]. Meanwhile, the MAPK signaling pathway and PI3K–Akt signaling pathway play an important role in glomerulosclerosis, and are the potential pathways of GN treatment [43]. There are also 19 oppositely expressed DIRGs that relate to T-cell differentiation or the MAPK pathway, but the genetic variation and molecular mechanisms in glomeruli and tubulointerstitium remain similar. The pivotal pathways and the genes involved are potential targets for understanding GN pathogenesis and exploring effective treatment.

The immune microenvironment is a central feature of inflammatory diseases, and ssGSEA was performed to further identify the unique immune infiltration mediated by multiple immune cells. The DCs and macrophages are increased in the glomeruli of GN. It is worth mentioning that DCs and macrophages constitute the main component of the immune system in the kidney, which could trigger the initiation of an immune response and lead to inflammation [44]. Moreover, there are more NK cells and monocytes in the tubulointerstitium, NK cells serve as an important source of IFN-γ in the fibrotic kidney, and there is a significant correlation between the number of NK cells and the histological severity of interstitial fibrosis [45]. In addition, monocytes are involved in the renal interstitial fibrosis through mutual activation with renal tubular epithelial cells [46]. Additionally, NKT and MAIT cells are significantly reduced in both glomeruli and tubulointerstitium. Their role in GN remains largely unclear, but there is no doubt that they are involved in the regulation of renal homeostasis through the secretion of pro-inflammatory and anti-inflammatory cytokines [47]. Furthermore, decreased iTreg and Th17 are observed in both glomeruli and tubulointerstitium. However, the abundance of Th17 in GN and control groups is numerically close.

iTregs can protect the kidney from injury in a variety of renal diseases by secreting a multitude of anti-inflammatory factors to suppress both adaptive and innate immune cells [48]. The defect in the quantity of Tregs make the inflammation more active in the kidney, which could be an important cause of GN [49]. Amplifying and inducting Tregs to restore immune homeostasis and tolerance may be a potential way to cure or control GN [50]. Current research show several discordant conclusions in the immune microenvironment of GN, but it is undeniable that there are many commonalities in different types of GN, and each of them has its unique characteristics of immune infiltration, which eventually leads to different pathological manifestations. Exploration of immune signatures are an important part in the realization of targeted and personalized therapy of GN.

As an efficient method to identify biomarkers, ML is widely used in the research of various diseases [51]. We screened seven genes (*ARG2, CSHL1, CX3CR1, LTF, LYZ, TMSB10, TRIM27*) based on regression analysis, and established a nomogram model for the diagnosis and prediction of GN. The calibration curve of the nomogram model indicates that it possesses a great prediction performance at a risk threshold from 0.1 to 0.9, and the high AUC value of the model in the validation set demonstrates its accuracy. However, it is difficult to distinguish multiple disease subtypes simultaneously using ML; for this reason, we tried to establish, for the first time, a DL model to identify characteristic genes of the eight GN subtypes. Twenty genetic biomarkers for glomeruli and tubulointerstitium of each GN subtype were screened out, and the DL models have an AUC value of more than 0.8 for each GN subtype. These genetic biomarkers are essential to define the pathogenesis of GN, develop targeted therapies, and offer a molecular perspective on the state of renal inflammation to search for key information for diagnosis and therapeutic management.

Regarding the limitations of this study, the clinical information is not rich enough to explore the biomarkers and molecular mechanism of disease severity and progression, and also further research is warranted to verify the results.

## 5. Conclusions

In this study, we screened out 274 common up-regulated or down-regulated DIRGs in the glomeruli and tubulointerstitium of GN and explored their functions. ssGSEA was also performed to identify the unique immune signatures of GN. In addition, seven genes (*ARG2, CSHL1, CX3CR1, LTF, LYZ, TMSB10, TRIM27*) were considered to be important biomarkers of GN diagnosis and prediction, and 20 DIRGs in glomeruli and tubulointerstitium of each subtype of GN were selected as characteristic genes based on DL.

## Figures and Tables

**Figure 1 biomolecules-12-01276-f001:**
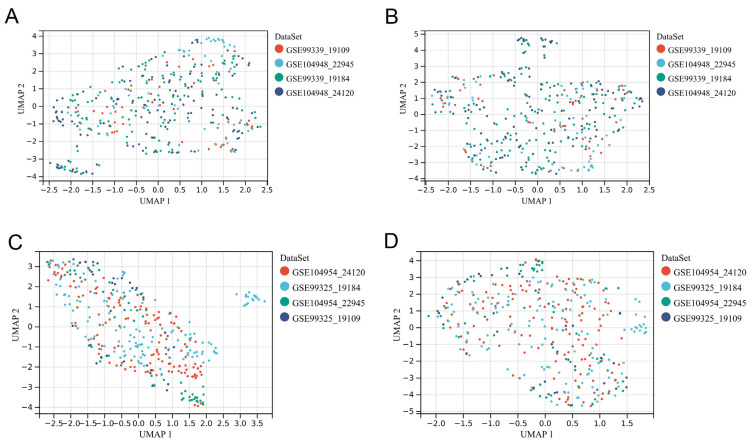
Merging datasets and removing batch effect. (**A**) The UMAP of sample distribution of each dataset of glomeruli before the removal of batch effect, samples from individual datasets are clustered separately, which indicates the existence of batch effect. (**B**) The UMAP of sample distribution of each dataset of glomeruli after the removal of batch effect, the samples of each datasets are clustered together, suggesting a good removal of batch effect. (**C**) The UMAP of sample distribution of each dataset of tubulointerstitium before the removal of batch effect, samples from individual datasets are clustered separately, which indicates the existence of batch effect. (**D**) The UMAP of sample distribution of each dataset of tubulointerstitium after the removal of batch effect, the samples of each datasets are clustered together, suggesting a good removal of batch effect.

**Figure 2 biomolecules-12-01276-f002:**
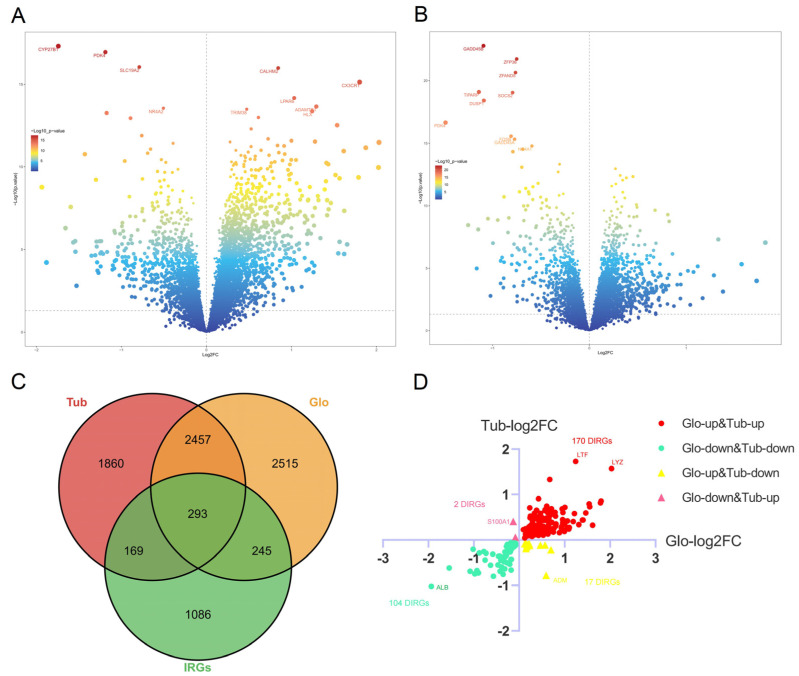
Identification of common DIRGs in data 1 and data 2. (**A**) Volcano map of all DEGs of data 1, there are 3474 up-regulated and 2036 down-regulated genes. (**B**) Volcano map of all DEGs of data 2, there are 2338 up-regulated and 2441 down-regulated genes. (**C**) Venn diagram of DEGs and IRGs, there are 274 common DIRGs in glomeruli and tubulointerstitium. (**D**) log2FC–log2FC plot of data 1 and data 2, there are 170 common up-regulated, 104 common down-regulated, and 19 oppositely expressed DIRGs.

**Figure 3 biomolecules-12-01276-f003:**
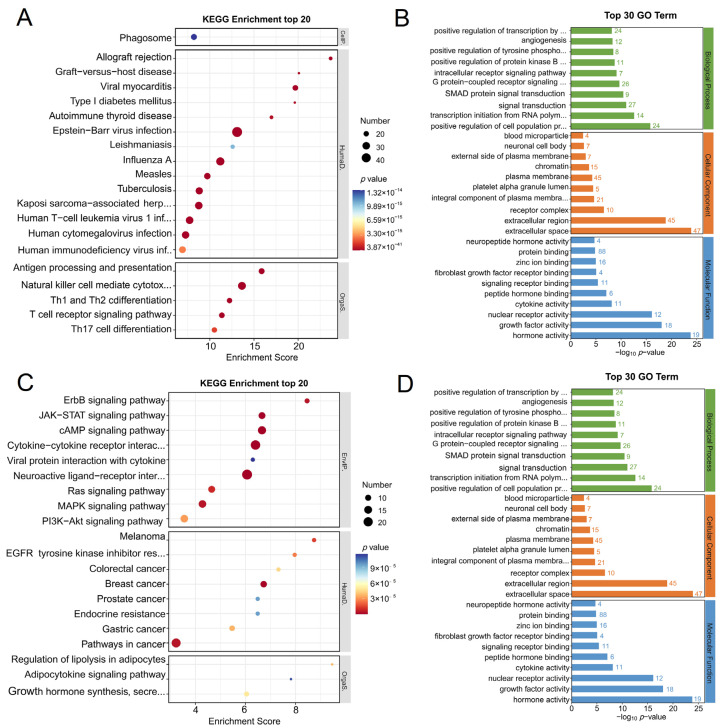
Enrichment analysis. (**A**) KEGG enrichment analysis of 170 common up-regulated DIRGs in data 1 and data 2. (**B**) GO enrichment analysis of 170 common up-regulated DIRGs in data 1 and data 2. (**C**) KEGG enrichment analysis of 104 common down-regulated DIRGs in data 1 and data 2. (**D**) GO enrichment analysis of 104 common down-regulated DIRGs in data 1 and data 2. The numbers next to the bars represent the number of genes enriched in the GO terms.

**Figure 4 biomolecules-12-01276-f004:**
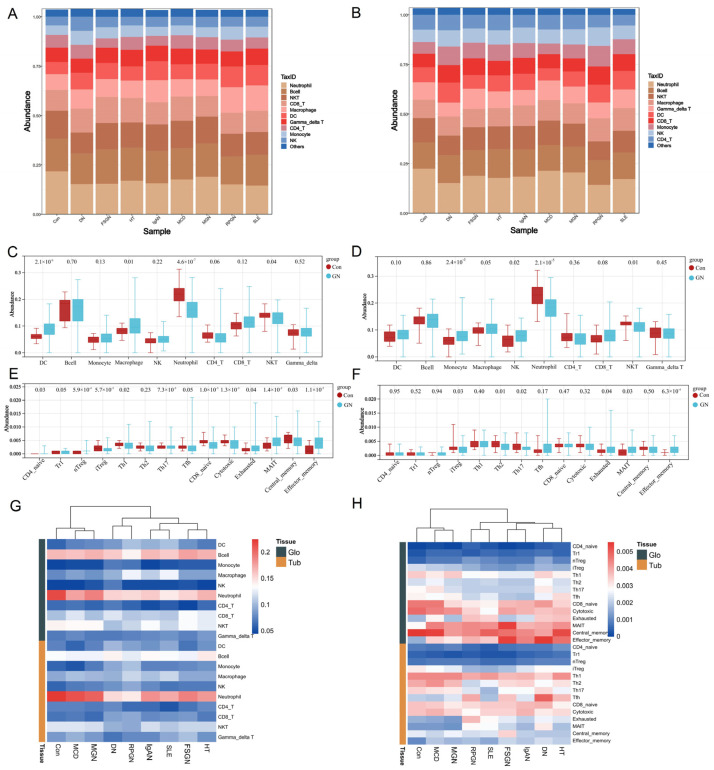
Immune signatures of the glomeruli and tubulointerstitium in GN. (**A**) Bar graph of 10 major types of immune cells in the glomeruli of 8 GN subtypes. (**B**) Bar graph of 10 major types of immune cells in the tubulointerstitium of 8 GN subtypes. (**C**) Boxplot of 10 major types of immune cells in the glomeruli of GN and healthy controls. (**D**) Boxplot of 10 major types of immune cells in the tubulointerstitium of GN and healthy controls. (**E**); Boxplot of 14 T-cell subtypes of immune cells in the glomeruli of GN and healthy controls. (**F**): Boxplot of 14 T-cell subtypes in the tubulointerstitium of GN and healthy controls. (**G**): Heatmap of 10 major types of immune cells in 8 GN subtypes and healthy controls. (**H**): Heatmap of 14 T-cell subtypes in 8 GN subtypes and healthy controls.

**Figure 5 biomolecules-12-01276-f005:**
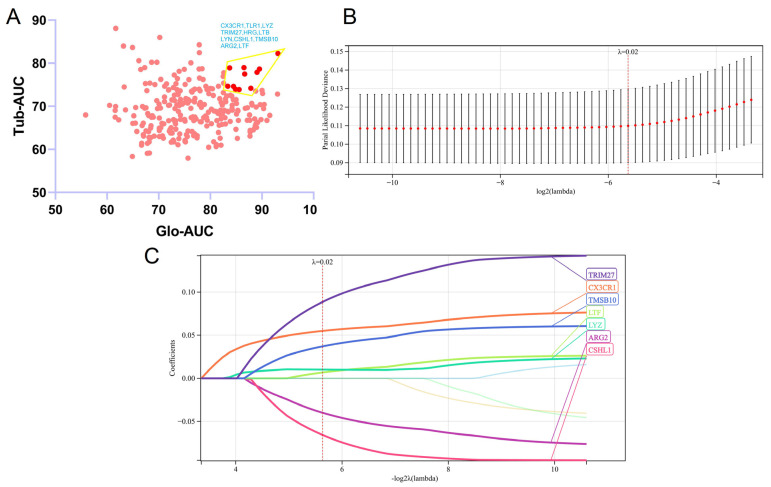
Gene screening and regression model establishment. (**A**) AUC−AUC plot of 274 common up-regulated or down-regulated DIRGs in data 1 and data 2, the DIRGs with top 20% AUC values both in glomeruli and tubulointerstitium are deep red. (**B**) The elastic net of 11 DIRGs in data 1. (**C**) Seven DIRGs were screened based on lambda = 0.02.

**Figure 6 biomolecules-12-01276-f006:**
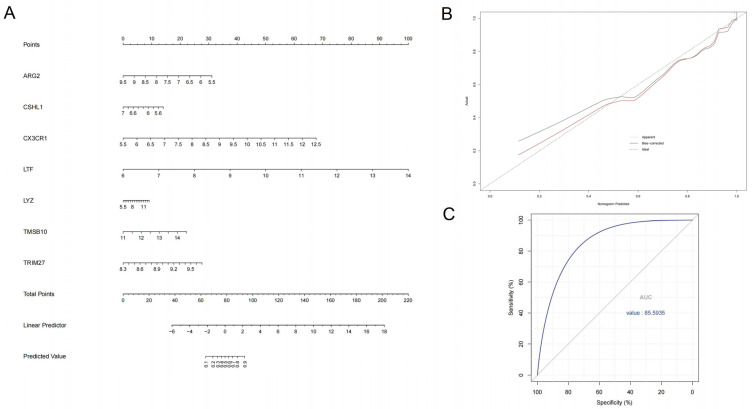
Construction and assessment of machine learning model. (**A**) Nomogram model for GN diagnosis, based on the 7 DIRGs (*ARG2, CSHL1, CX3CR1, LTF, LYZ, TMSB10, TRIM27*). (**B**) Calibration curve to evaluate the nomogram model. The actual GN risk and the predicted risk are very close. (**C**) The ROC curve to assess the nomogram model. The AUC value of the nomogram model in data 2 is 0.855935.

**Figure 7 biomolecules-12-01276-f007:**
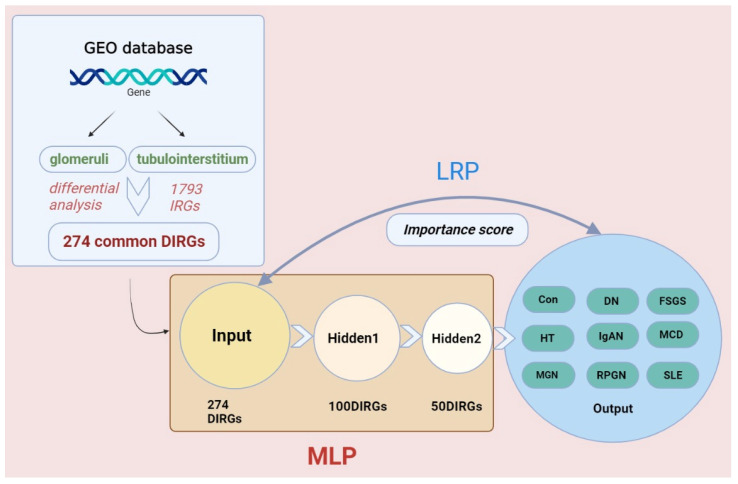
Workflow of the deep learning. Firstly, we downloaded microarray data from GEO database, screened candidate genes, and constructed MLP network. Then, LRP algorithm was used to calculate the characteristic genes of each GN subtypes.

**Figure 8 biomolecules-12-01276-f008:**
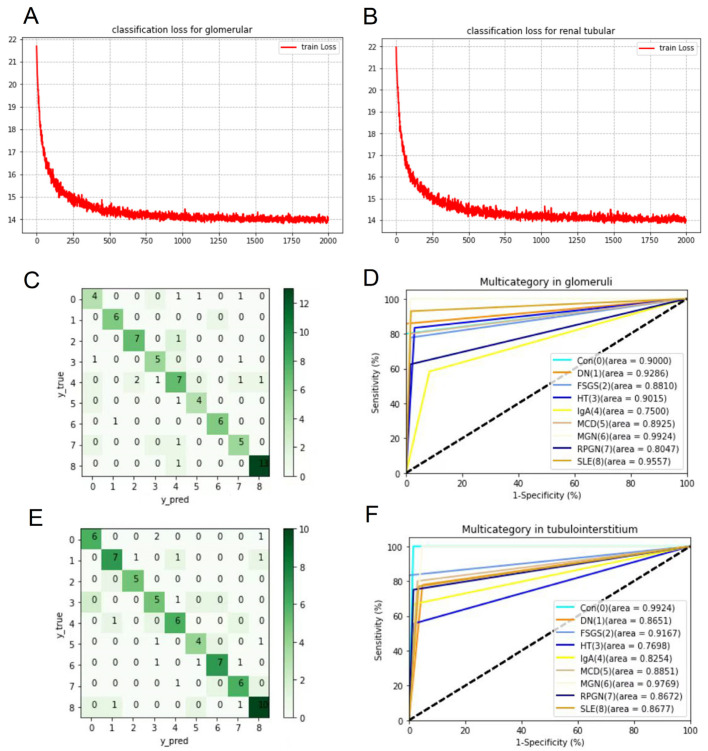
The loss curves, confusion matrix, and ROC curves of deep learning. (**A**) The loss curve of 274 candidate DIRGs based on data 1. (**B**) The loss curve of 274 candidate DIRGs based on data 2. (**C**) The confusion matrix of 274 candidate DIRGs based on data 1. (**D**) ROC curves of characteristic genes for each GN subtype in data 1. (**E**) The confusion matrix of 274 candidate DIRGs based on data 2. (**F**) ROC curves of characteristic genes for each GN subtype in data 2.

**Table 1 biomolecules-12-01276-t001:** The information of the datasets used for analysis.

Number	Platform	Tissue	Data Sources
GSE99339	GPL19109	Glomeruli	Shved, Natallia et al. [13]
	GPL19184	Glomeruli	
GSE104948	GPL22945	Glomeruli	Grayson, Peter C et al. [14]
	GPL24120	Glomeruli	
GSE99325	GPL19109	Tubulointerstitium	Shved, Natallia et al. [13]
	GPL19184	Tubulointerstitium	
GSE104954	GPL22945	Tubulointerstitium	Grayson, Peter C et al. [14]
	GPL24120	Tubulointerstitium	

**Table 2 biomolecules-12-01276-t002:** The information of all samples in data 1 and data 2.

Groups	The Number of Samples	
	Data 1 (Glomeruli)	Data 2 (Tubulointerstitium)
Control	21	25
Diabetic nephropathy (DN)	26	35
Focal and segmental glomerulosclerosis (FSGS)	40	25
Hypertensive nephropathy (HT)	33	40
IgA nephropathy (IgAN)	53	49
Minimal change disease (MCD)	27	25
Membranous glomerulonephritis (MGN)	42	36
Rapidly progressive glomerulonephritis (RPGN)	45	42
Lupus nephritis (LN)	62	62
Total	349	339

**Table 3 biomolecules-12-01276-t003:** The top five characteristic genes of eight GN subtypes.

Subtypes of GN	Glomeruli	Tubulointerstitium
DN	AVPR1A, GDF9, SEMA6C, ADA2, SSTR2	PLSCR1, CXCL8, TRIM22, CXCL1, PLXND1
FSGS	NFYA, MDK, BRD8, ADA2, JAK1	ADA2, VEGFC, NPY, ZAP70, IFNGR2
HT	TYMP, GRN, AMBN, CRIM1, SHC3	NR4A3, NR4A1, S100A8, ADIPOR2, SEMA6C
IgAN	INSR, PSMD7, GRN, TYMP, GDF9	MAP2K1, CSF1R, PTPN6, ZAP70, CD86
MCD	PSMD7, GRN, TNFRSF11B, TYMP, GIPR	MDK, PPARG, NFYA, CRABP1, ZAP70
MGN	NFKBIE, IL32, AVPR1A, PSMD7, CCL25	ZAP70, VEGFC, MAP2K1, GDF15, PPARG
RPGN	PAK4, CALCA, C3, C3AR1, ITGAL	OAS1, MAP2K1, GDNF, GDF2, FGF
SLE	GRN, B2M, DDX58, EIF2AK2, PTGDS	ADIPOR2, OAS1, EIF2AK2, MX1, PLSCR1

## Data Availability

The raw data supporting the conclusions of this article will be made available in GEO database. All data generated or analyzed during this study are included in this article and its Appendix A.

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
