# Peer review of "Identification of Unique Genetic Biomarkers of Various Subtypes of Glomerulonephritis Using Machine Learning and Deep Learning"

_biomolecules, 2022, doi:10.3390/biom12091276_

Round 1

Reviewer 1 Report

In this paper, a nomogram model was constructed to predict glomerulonephritis based on 7 DIRGs, and 20 DIRGs in glomeruli and tubulointerstitium of each subtype of glomerulonephritis. The paper is well written. The conclusion is pretty supported by the data and various methods. I would like to recommend publishing the manuscript in the journal after the authors proofread the manuscript and might fix some typos.

  1. The distribution of the data that used in the manuscript is needed to plot, like histogram. It would be better to use this instead of Panels A&B in Fig.2
  2. The font of Figs. 3&4 is too small to read.
  3. The panel C in Fig. 5 is not very clear. More colors would be used to distinguish the different trajectories. Moreover, what's the lambda on axis label?
  4. what's the machine learning model did you take in 3.5? more detailed information about network structure are needed. 
  5. In Fig. 7, the author claimed it is a deep learning but the only two hidden layers. A deep learning model would be interesting to see if the accuracy can be further improved.
  6. Fig. 8 shows the ROC curves. The loss curves are also needed to demonstrate the success of training neural networks.

Author Response

We are greatly indebted to you, for you can spare your precious time to review our paper, every word of your letter is very precious to us, We have read your questions and suggestions carefully, and modifications have been made for get your approval. If you have any other questions, please feel free to contact us, We hope to have more communication with you so that we can improve better. Looking forward to your reply. Thank you again for taking time out of your busy schedule to comments our article.

Answers

  1. The distribution of the data that used in the manuscript is needed to plot, like histogram. It would be better to use this instead of Panels A&B in Fig.2.

Thank you for this remark, we have plotted the data distribution bar chart as you suggested, which was showed in Supplementary figure1, besides, we wanted to use the volcano map to show the differentially expressed genes, thus, panels A&B in Fig.2 are retained.

  1. The font of Figs. 3&4 is too small to read.

Thanks for your note. We have adjusted the font in Fig.3 and 4 to make them more readable.

  1. The panel C in Fig. 5 is not very clear. More colors would be used to distinguish the different trajectories. Moreover, what's the lambda on axis label?

Thanks for this valuable advice. We have plotted the panel C in Fig.5 using richer colors, besides, the “lambda” means “λ” , and the corresponding supplementary explanations have been added in the text and Figure.

  1. what's the machine learning model did you take in 3.5? more detailed information about network structure are needed.

Thanks for this this crucial question. The machine learning model we took in 3.5 was logistic regression model, and we have made revisions to the methods section to include this detail.

  1. In Fig. 7, the author claimed it is a deep learning but the only two hidden layers. A deep learning model would be interesting to see if the accuracy can be further improved.

Thanks for this meaningful comment. In general, neural networks consist of many hidden layers, and to improve the accuracy of the DL model, we previously screened the genes used in the model, which makes our neural network end up with only two hidden layers.

  1. 8 shows the ROC curves. The loss curves are also needed to demonstrate the success of training neural networks.

Thanks for this note. We have drawn the loss curves according to your suggestion and added them to the new Fig.8.

Reviewer 2 Report

The study described herein applied AI on publicly available gene expression datasets from GN. The authors detected 274 DEG that are shared between the glomeruli and tubulointerstitium and could represent unique immune expression signature. Moreover, using ML the authors identified a set of 7 genes as potential biomarkers and additional 20 genes as characteristic for each GN subtype in the studied tissue.

Overall, the study is clear, concise and the claims are supported by the data.

However, there are some minor comments that if addressed will improve the manuscript.

1) Why data set1 was set as the training set? Is there certain parameters that should be taken into account when selecting a training set? It would be interesting to know if the analysis will give the same results if data set 2 was used as a training set.

2) Lines 144,145: ” …and we scored the importance of each gene in each subtype of GN. Finally, the unique biomarkers of each subtype of GN were screened out according to the score of DIRGs.” What is the cutoff value that was applied on the DIRG score for the screening process and what does it indicate? Is it fold change?

3) Figure1: the datasets appear to be fairly homogeneous as there is not an obvious batch effect and application of corrective methods did not make much of a difference on the data distribution, can the authors comments on this please.

4) The number of DIRGs that are shared between the two tissue is not consistent throughout the manuscript, in some occasions the indicated number is 273 and other 274.

5) Figure 3 (B,D): It would be more informative if the authors could include the number of queried genes (of the 274 DIRG)  that came up in each category

6) Figure 4

-(C-F): does each box represent number of DEG in each cell type?

- ( C ) NK cells have a comparable level in both Con and GN, can the authors revise this

- what type of statistical test was applied? And why symbols of significance are not indicated in the plot.

7) Discussion: it would enrich the discussion if the authors can compare their findings to other studies on GN or any other kidney dysfunction where a similar approach was applied to identify biomarkers

Example recent studies, doi.org/10.1186/s12882-022-02779-7,  doi: 10.3390/diagnostics11111983, doi.org/10.1016/j.bj.2021.08.011, doi.org/10.1007/s40620-021-01221-9

Minor corrections

 1) DIRGs definition should be consistent throughout (differentially expressed immune-related genes)

2) Table 1: if the numbers (13), (14) adjacent to the dataset ref refer to the source study, it would be better to add a column in the table for references.

 3) UMAP is misspelled in Figure 1 caption.

4) the labels in Figure 4 (A,B) X-axis are not legible

5) Human genes symbols are usually italicized

6) Figure 6 legend: the number of DIRGs upon the nomogram model was built should be  7 instead of 6

Author Response

We are greatly indebted to you, for you can spare your precious time to review our paper,every word of your letter is very precious to us, We have read your questions and suggestions carefully, and modifications have been made for get your approval. If you have any other questions, please feel free to contact us, We hope to have more communication with you so that we can improve better, looking forward to your reply. Thank you again for taking time out of your busy schedule to comments our article.

1) Why data set1 was set as the training set? Is there certain parameters that should be taken into account when selecting a training set? It would be interesting to know if the analysis will give the same results if data set 2 was used as a training set.

Thank you for this very important question. As the data in data 1 are more evenly distributed, and the main features of GN are in the glomeruli, they were used as training set. We have included the explanation of this choice in the current revision of manuscript. Regarding data 2, if it was used as a training set, the results might be different, but they would be less accurate.

2) Lines 144,145: ” …and we scored the importance of each gene in each subtype of GN. Finally, the unique biomarkers of each subtype of GN were screened out according to the score of DIRGs.” What is the cutoff value that was applied on the DIRG score for the screening process and what does it indicate? Is it fold change?

The score here refers to the importance score of the gene for the GN subtypes calculated by the LRP function, and each input gene will receive a score, and we finally screened the top 20 genes as feature genes and plotted ROC curves to test their accuracy.

3) Figure1: the datasets appear to be fairly homogeneous as there is not an obvious batch effect and application of corrective methods did not make much of a difference on the data distribution, can the authors comments on this please.

Preliminary data processing is crucial to the success or failure of the whole research. Although there is no significant difference in the sample distribution among different data sets as a whole, there are still some samples that need to be corrected. Therefore, in order to ensure the accuracy of the analysis, we removed the batch effect on all data sets.

4) The number of DIRGs that are shared between the two tissue is not consistent throughout the manuscript, in some occasions the indicated number is 273 and other 274.

Thanks for pointing this out. We have reviewed the full text and made revisions accordingly.

5) Figure 3 (B,D): It would be more informative if the authors could include the number of queried genes (of the 274 DIRG) that came up in each category

We have added a note of the number of genes next to the bars in the Figure 3 (B,D) based on your suggestion.

6) Figure 4-(C-F): does each box represent number of DEG in each cell type?- ( C ) NK cells have a comparable level in both Con and GN, can the authors revise this- what type of statistical test was applied? And why symbols of significance are not indicated in the plot.

Thanks for this crucial question. Firstly, each box in Figure 4(C-F) represents the abundance of each cell type. To avoid misunderstanding, we modified the Figure 4(C-F) by changing the “Expression” on the Y-axis to “Abundance”. Then, NK cells have a comparable level in both Con and GN, but NKT cells differed significantly, and more description was added in the corresponding section. Additionally, because we are more used to using specific P-values to represent the results of differential analysis, we did not use significance symbols in our manuscript, and we have added a note that “P values less than 0.05 are significant” in the revised manuscript.

7) Discussion: it would enrich the discussion if the authors can compare their findings to other studies on GN or any other kidney dysfunction where a similar approach was applied to identify biomarkers.

We've been thinking to compare the characteristic genes of each GN subtype to previous studies, and discuss their specific function, and how they participate in the pathogenesis of the disease. However, considering the space limitations of the article, we intend to make in-depth discussion and validation for the results of this manuscript in the next phase of the research.

Minor corrections

  1. We have modified the DIRGs in the article to “differentially expressed immune-related genes”.
  2. We have added a column in the table for references13 and 14.
  3. All misspellings have been corrected.
  4. The font in the Figure 4(AB) has been readjusted to make it clearer.
  5. The Human genes symbols have beenitalicized in the revised manuscript.
  6. The number of DIRGs in Figure 6 legendhas been modified to “7”.